Effects and potential mechanisms of iron metabolism on vascular calcification

Wang Hongyu 1 2 3
Song Yanqiu 1 2 3
Qin Qin cardioqq@126.com 1 2 3 4
1 Chest Hospital, Tianjin University , Tianjin , China
2 Tianjin Institute of Cardiovascular Disease, Tianjin Chest Hospital , Tianjin , China
3 Tianjin Key Laboratory of Cardiovascular Emergency and Critical Care, Tianjin Municipal Science and Technology Bureau , Tianjin , China
4 Department of Cardiology, Tianjin Chest Hospital , Tianjin , China
Uversky Vladimir
Electronic publication date: 2025 Dec 8
Publication date: 2025
Volume: 13
Electronic Location ID: e20392
Received 2025 Jul 18; Accepted 2025 Oct 24
Copyright: ©2025 Wang et al.
Copyright year: 2025
Copyright holder: Wang et al.
License: This is an open access article distributed under the terms of the Creative Commons Attribution License, which permits unrestricted use, distribution, reproduction and adaptation in any medium and for any purpose provided that it is properly attributed. For attribution, the original author(s), title, publication source (PeerJ) and either DOI or URL of the article must be cited.
License URL: https://creativecommons.org/licenses/by/4.0/

Keywords: Iron, Vascular calcification, Calcification treatment

Funding: Tianjin Health Science and Technology Project TJWJ2022QN069 Tianjin Key Medical Discipline Construction Project TJYXZDXK-3-017B Tianjin Chest Hospital Scientific Research Fund 2018XKZ10 This research was funded by Tianjin Health Science and Technology Project (TJWJ2022QN069), Tianjin Key Medical Discipline Construction Project (TJYXZDXK-3-017B) and Tianjin Chest Hospital Scientific Research Fund (2018XKZ10). The funders had no role in study design, data collection and analysis, decision to publish, or preparation of the manuscript.

==============================
Vascular calcification (VC) is a prevalent pathological manifestation of cardiovascular and cerebrovascular diseases, and is an active, multifactor-regulated pathological process. Iron is an essential metal that maintains cellular and body functions, and its metabolic homeostasis plays a complex and crucial dual role in the development of VC. This study, based on a comprehensive analysis of numerous studies, revealed that the effect of iron on VC has a significant dose-dependent relationship: physiological concentrations or moderate amounts of iron exert protective effects by enhancing antioxidant defenses, thereby inhibiting the osteogenic phenotype transformation and apoptosis of vascular smooth muscle cells; conversely, iron overload strongly drives VC by inducing oxidative stress, ferroptosis, and pro-inflammatory responses. These findings highlight the importance of maintaining iron homeostasis. Intervention strategies targeting iron metabolism (such as iron-based phosphate binders to correct iron deficiency and iron chelators to alleviate iron overload) have potential clinical value for the prevention and treatment of VC. In summary, this review provides a novel perspective on the diagnosis and treatment of VC, and future studies need to further explore its mechanisms and conduct rigorous clinical validation to manage iron metabolism as a novel approach for personalized prevention and treatment of VC.

Introduction

Normal physiological calcification is essential for bone formation, whereas ectopic calcification occurs in blood vessels, cartilage, and soft tissues. Ectopic calcification is affected by a variety of factors, including hormonal imbalances, oxidative stress, cellular nucleotide metabolism (Opdebeeck et al., 2020), cell–cell interactions, the extracellular matrix, and mechanical forces. Ectopic calcification of blood vessels is known as vascular calcification (VC). VC is a common pathological process observed in various vascular diseases (Zhao et al., 2024). Studies have consistently shown that both intimal and medial calcifications increase the risk of adverse events. Coronary artery calcification has been demonstrated to elevate the risk of all-cause mortality and fatal myocardial infarction (Bourantas et al., 2014). Moreover, calcification of the abdominal aorta, femoral artery, and mammary artery increases the risk of all-cause and cardiovascular mortalities (Niu et al., 2019).

Iron is an essential metallic element that is crucial for maintaining normal cellular function in mammals, where it plays a vital role in various physiological processes, including oxygen transport, erythropoiesis, energy metabolism, and DNA synthesis (Muckenthaler et al., 2017). Iron deficiency and overload can threaten human health and lead to serious diseases (Dev & Babitt, 2017). The dual nature of iron requires strict regulation at systemic and cellular levels, making iron homeostasis a critical and highly regulated process for maintaining human health (Bogdan et al., 2016). The intricate balance of iron homeostasis is maintained by a complex interplay of factors, such as iron absorption, storage, and metabolism. Iron levels strongly correlate with calcium content in cells and tissues. For example, a significant correlation between cardiac iron uptake and calcium storage has been observed in a mouse model (Otto-Duessel, Brewer & Wood, 2011). Furthermore, elevated hepcidin levels are significantly associated with atherosclerosis and cardiovascular events in clinical studies (Afsar et al., 2025). Nonetheless, the mechanism of action of iron in VC is complex and inconclusive.

Conventional knowledge has long held that VC is a passive, unavoidable, end-stage process. However, recent studies have shown that VC is a preventable and modifiable active process, analogous to bone development and cartilage formation. Despite this, the underlying mechanisms of VC development is still unclear, and as a result, no therapeutic agents or methods have been found to be effective in halting or reversing VC (Cao et al., 2013). Previously, VC was believed to favor local plaque stability. However, subsequent studies have increasingly demonstrated that coronary artery calcification is associated with reduced target vessel compliance, which is unfavorable for local revascularization therapy. This greatly increases the difficulty of percutaneous coronary intervention, reduces the success rate of the procedure, and increases the incidence of post-procedural complications, which can have a significant impact on prognosis and impose a significant burden on patients’ healthcare resources. This review comprehensively analyzed a large number of studies on the effect of iron on VC, and the iron metabolism pathway guiding the treatment of VC, with the aim of elucidating the effect and potential mechanism of iron metabolism on VC and providing a novel perspective for the diagnosis and treatment of VC.

Intended population

The primary population for this review is researchers studying the mechanisms of VC or the role of iron metabolism, as well as physicians working with calcification-related diseases. It synthesizes the understanding of the role of iron metabolism in VC and current therapeutic strategies for VC treatment guided by the iron metabolism pathway. The main objectives of this study were to elucidate the effects and underlying mechanisms of iron metabolism in VC and provide new perspectives for the diagnosis and treatment of VC.

Survey Methodology

Literature searches were conducted using the PubMed, Google Scholar, and Web of Science databases. Keywords used included “vascular calcification,” “calcification mechanism,” “vascular smooth muscle cells and calcification,” “calcification regulation,” “inflammation and calcification,” “apoptosis and calcification,” “autophagy and calcification,” “miRNA and calcification,” “iron metabolism,” “iron and calcification,” “calcification treatment,” and “iron therapy.” Research and review articles published in English language were included in this review. Letters and case report to the editor and case report were excluded. A total of 52 articles published between 2001 and 2025 were included in this review.

VC

Accumulating evidence suggests that VC is a tightly regulated process involving a balance between VC induction and inhibition (Zoccali & London, 2015). Cardiovascular calcification can develop in different areas, and depending on the location of the calcification, it can be classified as intimal calcification, medial calcification, adventitial calcification, calciphylaxis, or valve calcification (Tóth, Balogh & Jeney, 2024). These distinct sites and forms of calcification are caused by different pathological processes that may interact with and influence each other. The mechanisms underlying VC are complex and still unclear. The previous understanding that VC is a passive process has been updated; it is now thought to be an active, cell-mediated, regulated, osteogenesis-like complex process (Nakahara et al., 2017).

The phenotypic switch of vascular smooth muscle cells (VSMCs) from the contractile to the osteogenic type is a key mechanism underlying VC (Liou et al., 2020). VSMCs are the major cell type in blood vessel walls and typically exhibit a contractile phenotype. Contractile VSMCs strongly express smooth muscle α-actin, smooth muscle 22α, smooth muscle myosin heavy chain, calponin, and smoothelin (Durham et al., 2018). Phenotypic transformation of VSMCs involves the downregulation of the aforementioned contractile protein expression, as well as the downregulation of the characteristic VSMC differentiation marker gene alpha-smooth muscle actin, and the upregulation of the typical osteoblast-specific gene runt-related transcription factor 2 (RUNX2) (Steitz et al., 2001; Wang et al., 2019; Wang et al., 2018). Additionally, intercellular communication between VSMCs occurs through stromal vesicles (Chen, O’Neill & Moe, 2018). Contractile VSMCs undergo phenotypic transformation after endocytosis of matrix vesicles secreted by osteogenic VSMCs, ultimately leading to VC.

Apoptosis also plays a crucial role in regulating calcification in VSMCs (Ciceri et al., 2016; O’Neill, Sigrist & McIntyre, 2010). In an in vitro human VSMC calcification model, apoptosis precedes calcium deposition. Apoptotic bodies and calcium-loaded vesicles from apoptotic cells serve as nucleation sites for the formation of calcific lesions, which promote the deposition of calcium phosphate crystals in the extracellular matrix and accelerate calcification (Proudfoot et al., 2001). Therefore, an increase in apoptosis is the main mechanism that promotes VC (Komuro et al., 2024). In contrast, VC can be significantly improved by inhibiting apoptosis (Yu et al., 2024). Additionally, autophagy is intimately linked with apoptosis, which is triggered when autophagy fails to reestablish a positive equilibrium for cell survival. Studies have also indicated that autophagy is an endogenous protective mechanism against VC, and that increasing autophagy levels can attenuate VC (Yu et al., 2024).

Recent studies have highlighted the roles of mitochondrial dysfunction and endoplasmic reticulum stress in VC pathogenesis (Chen et al., 2025; Rao et al., 2022; Wang et al., 2020). The mitochondria can directly or indirectly regulate VC through oxidative stress damage, autophagy, mitochondrial autophagy, and apoptosis. Additionally, damage to mitochondrial DNA by adverse factors may damage the structure and function of the mitochondrial electron transport chains, ultimately leading to VC. Endoplasmic reticulum stress promotes VC development by promoting osteogenic transformation, inflammation, autophagy, and apoptosis in VSMCs and endothelial cells. Furthermore, a multitude of biological processes occur in calcific disease, including miRNA regulation, matrix degradation and remodeling, cellular senescence, abnormal lipid and mineral metabolism, loss of mineralization inhibition, and impaired mineral resorption (Cozzolino et al., 2019; Pescatore, Gamarra & Liberman, 2019; Rogers & Aikawa, 2018).

Effect of iron on VC

Extensive cellular and animal experimental evidence suggests that iron plays a dual role in promoting and inhibiting VC, a contradiction indicating a complex mechanism of action (Table 1).

Table 1 Summary analysis of the dual effect of iron on vascular calcification (VC).

Analysis dimensions	Inhibitory effect (moderate iron)	Promoting effect (iron overload)	
Core mechanism	Antioxidation and cell protection	Ferroptosis and oxidative damage	
Overview of the mechanism	Removes ROS;
Maintains the contractile phenotype of VSMCs;
Inhibits osteogenic differentiation.	The large accumulation of ROS and lipid peroxides;
Cells undergoing ferroptosis release vesicles rich in calcium and phosphate, providing ideal nucleation sites for calcification.	
Action type
(Empirical evidence)	In vitro:
Induced ferritin and its ferroxidase activity;
Induce autophagy and inhibit apoptosis;
Regulate the levels of exosomal miRNA;
Downregulate the expression of osteoblast markers (such as BMP2).	In vitro:
Inducing ferroptosis and oxidative damage exacerbates local inflammation;
Enhance the expression of regulatory factors that promote calcification (BMP2, RUNX2, RANKL, etc.);
Impair the function of calcification inhibitors (such as Gla proteins and fetuin-A).	
In vivo:
Iron-based phosphate binders improve iron status and anaemia, while controlling blood phosphate levels;
Prevent the dissolution of elastic fibres and the deposition of collagen;
Inhibite apoptosis by restoring the Gas6/Axl pathway, protecting cells from high phosphate-induced calcification.	In vivo:
NTBI accumulates in the arterial walls, leading to oxidative stress, cell apoptosis, and inflammation;
Cause ferroptosis, iron chelating agents or iron death inhibitors can inhibit its pro-calcification effect;
Promote lipid oxidation (such as the formation of oxLDL), foam cell formation, and EC activation;
Induce hypercalcaemia and hyperphosphataemia.	
Deciding factor	Follow the dose–response relationship:
Physiological concentrations or slight increases in iron provide a protective effect, inhibiting VC;
Once a certain critical threshold is exceeded, oxidative damage occurs, leading to ferroptosis and promoting VC.	
The critical threshold may vary due to individual conditions and the cellular microenvironment, with complex interactions collectively determining the final cell fate and calcification outcomes.	
Notes.

Abbreviations ROS reactive oxygen species

VSMCs vascular smooth muscle cells

BMP2 bone morphogenetic protein 2

RUNX2 runt-related transcription factor 2

RANKL receptor activator of nuclear factor kappa-B ligand

NTBI non-transferrin bound iron

EC endothelial cell

Inhibitory effect of iron: antioxidant and cellular protective effects

Iron exerts a protective effect through its antioxidant properties, which inhibit VC initiation and progression. Excessive generation of reactive oxygen species (ROS) can induce osteoblastic differentiation of VSMCs, promote apoptosis, and release calcifying vesicles. Iron is an essential cofactor for various antioxidant enzymes such as catalase and glutathione peroxidase. Adequate iron levels ensure the activity of these enzymes, effectively clearing ROS, reducing oxidative damage, and maintaining the contractile phenotype of VSMCs.

Several studies have supported this hypothesis. In in vitro experiments, iron inhibited the osteogenic differentiation of VSMCs by inducing ferritin and its ferroxidase activity, and inhibited apoptosis while inducing autophagy (Ciceri & Cozzolino, 2020; Wickens et al., 2022). Iron can also inhibit calcification through other mechanisms. Iron can inhibit the osteochondrogenic transformation of VSMCs by regulating the levels of exosomal miRNAs and downregulating the expression of osteoblast markers, such as bone morphogenetic protein 2 (BMP2) (Ciceri et al., 2019a), thereby preventing phosphate-induced VC (Ciceri et al., 2019a). Second, iron inhibits apoptosis by restoring the Gas6/Axl pathway, thereby protecting cells from high phosphate-induced calcification (Ciceri et al., 2019b). Notably, when calcification has already occurred, the addition of iron can delay and block apoptosis progression induced by high phosphate levels, reversing the production of calcification particles and osteochondrogenesis in the extracellular matrix, thereby blocking calcification.

In vivo experimental studies corroborated these results. Iron-based phosphate binders (such as iron citrate and sucrose iron hydroxide) can improve iron status and anemia while controlling blood phosphate levels, and have been shown to directly inhibit VC in animal models (Ciceri & Cozzolino, 2020). Additional in vivo studies have found that in the aortic ring, iron prevents the dissolution of elastic fibers and collagen deposition induced by high phosphorus in the extracellular matrix (Ciceri et al., 2019a), thereby blocking calcification. Additionally, iron can negatively regulate calcification by reducing the systemic levels of fibroblast growth factor 23 (Schouten et al., 2025). Although excess iron increases the levels of ROS, which can trigger molecular mechanisms leading to VC, iron deposition induces protective molecules, such as ferritin and heme oxygenase 1, to inhibit this process.

In summary, an adequate amount of iron can exert a protective effect against calcification by enhancing the antioxidant defense system, counteracting the oxidative stress signals that promote calcification, and stabilizing the contracted phenotype of VSMCs.

Promoting role of iron: ferroptosis and cellular stress damage

In stark contrast to the above findings, another series of robust studies indicated that iron is also a significant risk factor for promoting VC, with the core mechanism revolving around iron overload and explosive generation of ROS. Ferroptosis is characterized by a large accumulation of iron-dependent lipid peroxides, leading to rupture of cell membranes. When VSMCs undergo ferroptosis, they release a large number of calcium- and phosphorus-rich vesicles, exposing nucleation sites, such as phosphatidylserine, thereby providing ideal nucleation sites for hydroxyapatite deposition, while also exacerbating local inflammation and further promoting the differentiation of surrounding VSMCs into osteoblasts, which accelerates calcification.

Iron overload can lead to ferroptosis, resulting in the failure of iron-dependent lipid peroxidation and antioxidant system (GPX4), causing lipid peroxidation and cell membrane rupture, as well as significant accumulation of ROS, which promotes the osteoblastic phenotype transformation of VSMCs and calcification (Li et al., 2022). Iron can also promote calcification through other pathways. Iron can enhance the expression of certain important regulators that promote calcification, including BMP2, RUNX2, MSH homolog 2, and receptor activator of nuclear factor kappa-B ligand (Hortells et al., 2017; Liou et al., 2020; Shao et al., 2005; Zhou et al., 2019), ultimately leading to the occurrence or progression of calcification. Excessive iron can also affect the function of calcification inhibitors (such as Gla proteins and fetuin-A), resulting in increased calcification (Wang et al., 2021).

Results of the in vivo study further confirmed the role of iron in promoting calcification. Animal experimental studies have found that in the case of iron overload, non-transferrin-bound iron can accumulate in the arterial walls of mice, particularly in VSMCs, leading to oxidative stress, cell apoptosis, and inflammation (Vinchi, 2021). Furthermore, iron overload can promote lipid oxidation (such as the formation of oxLDL), foam cell formation, and endothelial cell activation, ultimately leading to calcification (Vinchi, 2021). Iron overload also impairs renal function, induces hypercalcemia and hyperphosphatemia, leads to aortic tissue calcification, and promotes VC by upregulating osteogenic differentiation factors and downregulating calcification inhibitors (Song et al., 2022). Additionally, iron overload can lead to ferroptosis, and the use of iron chelators or ferroptosis inhibitor, ferrostatin-1, can suppress ferroptosis and activate antioxidant pathways, thereby inhibiting calcification (Li et al., 2022).

In summary, iron overload induces ferroptosis, leading to lethal membrane lipid peroxidation in VSMCs. These cells not only lose the ability to maintain vascular elasticity but also create an environment that promotes the mineralization process, thereby driving VC strongly.

Analysis of the dual effects of iron metabolism on VC

The effects of iron may exhibit a dose–response relationship. Physiological concentrations or slightly increased iron may act as a cofactor for the activation of antioxidant pathways, thereby providing protective effects; however, once it exceeds a certain critical threshold, it can produce a large amount of ROS through the Fenton reaction, triggering ferroptosis and leading to calcification, among other effects. This critical threshold may vary, depending on an individual’s health status and the cellular microenvironment (e.g., phosphate levels). Additionally, various cell types in the vascular wall (such as VSMCs, endothelial cells, and fibroblasts) may have different sensitivities to iron concentrations and respond differently, whereas inflammatory factors, phosphate, calcium ion levels, and other components in the microenvironment may engage in crosstalk with iron signaling. These complex interactions collectively determine the final cell fate and calcification outcomes (Fig. 1).

Figure 1 Effects and potential mechanisms of iron metabolism on vascular calcification.

Abbreviations: DFO, desferrioxamine.

Interventions targeting iron metabolism as a potential therapeutic strategy in VC

VC is an important independent predictor of the incidence and mortality of cardiovascular diseases, and is commonly observed in patients with chronic kidney disease (CKD), diabetes, and in older adults. A basic study has revealed the dual complex role of iron in VC; however, as an essential trace element for the human body, the clinical relevance of iron and its potential treatment strategies in relation to VC still need to be further explored through clinical studies. This section will analyze and discuss the clinical evidence linking iron metabolism disturbances to VC, as well as potential treatment strategies based on iron regulation (Table 2).

Table 2 Clinical association and treatment strategies of iron metabolism disorders and vascular calcification (VC).

Core dimension	Core viewpoint	Essential evidence/mechanism	
Core contradiction	Iron homeostasis imbalance is a key risk factor for VC, with both iron deficiency and iron overload posing threats.	Clinical evidence confirms the dual role of iron in various metabolic processes.	
Clinical evidence	Iron deficiency promotes the progression of VC, especially in the CKD population.	Observational studies have confirmed that iron deficiency indicators are independently associated with the extent of coronary calcification, stemming from a weakened antioxidant defence system.	
Excessive iron loading also exacerbates VC risk.	Iron overload induces ferroptosis and inflammation through the Fenton reaction, creating a pathological environment for calcification.	
Treatment strategy	The essence lies in bidirectional precise adjustment: make up for deficiencies and eliminate excess.	The treatment paradigm is expanding from correcting anaemia to the field of cardiovascular protection.	
Iron supplementation strategy: intraven-
ous iron agents and iron-based phosphate binders are the mainstay.	The former may improve heart function and indirectly combat calcification; the latter has dual benefits of lowering phosphorus (a key driver of VC) and supplementing iron.	
Iron removal strategy: iron chelators can be used for iron overload related VC.	It reduces oxidative stress by lowering iron load, and basic research has shown that it can directly alleviate vascular smooth muscle cell calcification.	
Future direction	Targeting ferroptosis and other downstream events is an emerging therapeutic paradigm.	Iron death inhibitors have demonstrated strong direct anti-calcification efficacy in preclinical studies.	
Personalised treatment is the ultimate goal.	There is a need to develop new biomarkers for the precise regulation of local vascular iron homeostasis.	
Notes.

Abbreviations CKD Chronic Kidney Disease

Clinical evidence of the effect iron on VC: contradictions and complexities

Clinical evidence suggests that the body’s iron status, whether deficiency or overload, may be associated with an increased risk of VC. This underscores the importance of iron metabolic homeostasis and further corroborates the dual-edge effect of iron on VC observed in basic studies.

Iron deficiency is commonly found in high-risk populations, such as those with CKD, and is associated with VC progression. This mechanism may be related to iron deficiency, which weakens the antioxidant defense capabilities of the body. Observational studies have shown that iron deficiency (low transferrin saturation) in patients with CKD is associated with increased coronary artery calcification scores and high mortality rates (Mizuiri et al., 2021). Clinical studies have found that patients with CKD often have anemia and iron deficiency, whereas VC is a major independent risk factor for cardiovascular events and all-cause mortality in patients with CKD, indicating a strong link between iron deficiency and VC development (Ciceri & Cozzolino, 2020).

In contrast, iron overload is associated with an increased risk of VC. Iron overload generates a large number of hydroxyl radicals through the Fenton reaction, triggering intense oxidative stress and ferroptosis, thereby accelerating calcification. A study exploring the pathophysiological role of the iron-regulating hormone hepcidin and its potential as a therapeutic target indicated that hepcidin is a core regulator of systemic iron homeostasis, and that its dysfunction (either deficiency or excess) is a fundamental cause of various iron-related diseases (Fung & Nemeth, 2013). Importantly, during iron overload, insufficient secretion of hepcidin can lead to severe oxidative stress and inflammation in cells, resulting in tissue damage. These processes create ideal conditions for pathological calcification (including VC) (Fung & Nemeth, 2013).

Therapeutic potential of iron regulation in clinical studies

Based on the above mechanisms, regulating iron metabolism may be a promising therapeutic strategy for interventions in VC. Traditionally, oral iron supplementation has been the simplest, most direct, and cost-effective method for treating iron deficiency. However, oral iron supplementation has some drawbacks, such as poor compliance, gastrointestinal side effects, and suboptimal treatment outcomes. In contrast, intravenous iron therapy has been shown to be more effective in patients who exhibit intolerance or an inadequate response to oral iron therapy (Ghafourian et al., 2020; Ning & Zeller, 2019). Therefore, most improvements in diseases caused by iron deficiency or iron metabolism disorders are currently achieved using intravenous iron preparations.

Currently, the most direct clinical evidence comes from the use of iron-based phosphate binders. Hyperphosphatemia is a key factor driving VC in patients with CKD. Ferric citrate and iron sucrose not only effectively reduce blood phosphate levels, but the iron they provide may also have a direct anti-calcification effect within a certain threshold (Lewis et al., 2015). VC is a major independent risk factor for cardiovascular events and all-cause mortality in patients with CKD. Iron deficiency is a common comorbidity of cardiovascular disease and heart failure (HF), and approximately 50% of patients with HF experience iron deficiency (with or without anemia). Studies have shown that iron-based phosphate binders (such as iron citrate and sucrose iron hydroxide) can control blood phosphate levels and improve iron status and anemia, thereby inhibiting VC (Macdougall et al., 2018). Additionally, clinical trials have indicated that intravenous ferric carboxymaltose (FCM) can improve HF and quality of life (Ciceri & Cozzolino, 2020). These iron-based phosphate binders have beneficial effects on CKD and HF, including alleviation of VC. A clinical trial assessing the impact of intravenous FCM on the health-related quality of life of hospitalized patients with acute HF due to iron deficiency also provides some insights (Jankowska et al., 2021). Although the primary endpoint of this study was quality of life rather than a direct investigation of calcification, it provides important clinical evidence for understanding the relationship between iron status and the cardiovascular system, including potential calcification processes. Intravenous FCM (Scott, 2018) can rapidly and significantly improve the symptoms and functional status of patients with HF. This improvement is likely due to iron-optimizing mitochondrial function and energy metabolism, which enhances cardiac function and systemic tissue oxygenation. A heart with improved function and reduced hypoxia may have a tissue environment less conducive to the progression of pathological calcification. Additionally, correcting iron deficiency may enhance the antioxidant capacity of cells, thereby alleviating tissue damage and inflammation driven by ROS, whereas oxidative stress and inflammation are the core mechanisms driving VC. Additionally, ferric desmosomaldehyde (FDI) can be used as a single high-dose rapid iron supplement and has been shown to be safe and well tolerated. It is worth mentioning that compared with FCM, the unstable iron levels released by FDI are lower (Emrich et al., 2020; Kassianides, Bodington & Bhandari, 2020), which can also make it a suitable therapeutic iron preparation.

However, iron chelators have a therapeutic potential for iron overload. Despite the lack of large-scale clinical trials for VC treatment, basic studies have indicated that iron chelators can alleviate high phosphate-induced calcification of VSMCs and VC in animal models. A clinical study discussing the application of two iron chelators, desferrioxamine (DFO) and deferiprone, in the treatment of chronic iron overload caused by transfusions found that iron overload was a strong promoting factor of pathological calcification. Iron chelators (such as DFO and deferiprone) effectively reduce iron levels in the body, reduce oxidative stress, and improve organ function (such as cardiac function), creating an internal environment unfavorable for calcification (Cappellini, Musallam & Taher, 2009). Therefore, the use of iron chelators to reduce iron load in iron overload disease has been shown to improve various outcomes, including cardiovascular complications, which also supports the view that reducing iron load may be beneficial in alleviating VC. New therapeutic targets, such as inhibitors of ferroptosis (like ferrostatin-1), have recently shown strong anti-calcification effects in preclinical studies, but their potential for clinical translation requires further exploration.

In summary, clinical observational evidence clearly reveals that maintaining iron metabolic balance and avoiding extreme conditions of iron deficiency or overload are crucial for bodily health. Targeted iron regulation strategies, such as using iron-based phosphate binders in patients with CKD having iron deficiency, or cautiously using iron chelators in patients with iron overload, expand iron management from its current focus on anemia to cardiovascular protection, particularly in the field of anti-calcification, as well as potential therapeutic benefits in delaying or even reversing VC progression (Fig. 1). However, further clinical trials are required to provide direct and conclusive evidence. Exploring individualized treatments by monitoring new biomarkers (such as those related to ferroptosis) or by precisely regulating local vascular iron homeostasis may provide promising new avenues for the prevention and treatment of this challenging issue of VC.

Conclusions

VC is associated with the development of life-threatening cardiovascular diseases, and is a common pathological manifestation of cardiovascular diseases. Iron is an important metallic element that maintains normal cellular function in mammals. Iron deficiency and overload can adversely affect cells and individuals, making maintenance of iron homeostasis crucial.

This study systematically reviewed numerous studies, establishing the central role and dual function of iron metabolism in VC (Fig. 1). Physiological concentrations or moderate amounts of iron can enhance cellular defense mechanisms by acting as a cofactor for antioxidant enzymes, inhibiting the osteogenic transformation and apoptosis of VSMCs, thereby exerting a protective effect. Conversely, iron overload induces oxidative stress and ferroptosis through the Fenton reaction, upregulating osteogenic gene expression and strongly driving VC. This reveals that VC is an active pathological process intricately regulated by multiple factors, including iron metabolism. Intervention strategies targeting iron metabolism (such as iron-based phosphate binders and iron chelators) have shown clear clinical translational potential.

Based on these conclusions, further studies have urgently required breakthroughs in multiple directions. First, in-depth mechanistic studies should be conducted to explore the interactive network between iron signaling and microenvironmental factors in specific pathological contexts. Second, clinical translation is of utmost importance, requiring the promotion of large-scale prospective studies to obtain hard endpoint evidence of the direct impact of iron regulation strategies on VC progression, and the active exploration of personalized treatment plans based on individual iron metabolic states. Finally, it is essential to identify and validate novel circulating biomarkers that can reflect local iron status and ferroptosis activity in blood vessels to enable early identification of high-risk groups for VC and monitor treatment responses. In summary, iron metabolism is a key and promising mechanism of VC regulation. By deepening our understanding of the underlying mechanisms and actively promoting clinical translation, management strategies targeting iron homeostasis are expected to open novel avenues for the prevention and treatment of this significant clinical challenge.

Additional Information and Declarations

Competing Interests

Author Contributions

Data Availability

The authors declare there are no competing interests.

Hongyu Wang conceived and designed the experiments, performed the experiments, analyzed the data, authored or reviewed drafts of the article, and approved the final draft.

Yanqiu Song performed the experiments, authored or reviewed drafts of the article, and approved the final draft.

Qin Qin conceived and designed the experiments, authored or reviewed drafts of the article, and approved the final draft.

The following information was supplied regarding data availability:

Raw data was not generated in this literature review.

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
