# Peer review of "Effects and potential mechanisms of iron metabolism on vascular calcification"

_PeerJ, doi:10.7717/peerj.20392_

## Round 0.1 · original submission · Major Revisions

· Academic Editor

Major Revisions

Please note that although one of the reviewers recommended rejection, in light of the timeliness of the topic of the relationship between iron and vascular calcification, I decided to give you an opportunity to prepare the revised manuscript. Please address concerns of both reviewers, reply to their queries, and amend the manuscript accordingly.

Reviewer 1 ·

Basic reporting

In this review manuscript, Wang et al. summarize the current understanding of iron metabolism and vascular calcification (VC). The manuscript is structured into three main sections: 1)” Vascular Calcification”; providing an overview of the mechanisms underlying VC, 2) “The correlation between iron levels and vascular calcification”; presenting in vitro and in vivo experimental data examining the causal relationship between iron and VC, and 3) “Treatment of vascular calcification based on iron metabolic pathways”; discussing clinical evidence on the connection between iron and VC, and exploring the potential of targeting iron metabolism. In clinical practice, both iron deficiency and VC are well recognized, particularly in patients with end-stage kidney disease. However, the causal relationship between these two conditions remains unclear. This review contributes valuable insights for readers of PeerJ by outlining the current state of knowledge. The following comments are intended to help improve the clarity, completeness, and impact of the manuscript.

1) In Chapter 1, “Vascular Calcification”, the role of apoptosis should be more strongly emphasized. Apoptotic bodies and matrix vesicles released from dying and/or living cells can act as nucleation sites for calcification, particularly in the context of VC and other pathological calcification processes. These vesicles provide a localized environment conducive to the precipitation of calcium phosphate crystals. It would be appropriate to address this in one of the early subheadings within the chapter.

2) Recent studies have highlighted the role of mitochondrial dysfunction and ER stress in the pathogenesis of VC (e.g., Wang P, J Transl Int Med 2020. PMID 32983930., Rao Z, Front Cardiovasc Med 2022. PMID 35783850.). Including a discussion of these emerging mechanisms in Chapter 1 would provide a more comprehensive overview.

3) (Page 5, line 166) The phrase “pro- and anti-calcification mediators within the arterial wall” requires further elaboration. A more detailed explanation with appropriate references (e.g., Nakahara T, JACC Cardiovasc Imaging 2017. PMID 28473100) would improve clarity for readers unfamiliar with these concepts.

4) Chapter 2, “The Correlation Between Iron Levels and Vascular Calcification”, presents substantial data from in vitro studies on the pro- and anti-calcific effects of iron. However, there is limited discussion of in vivo evidence, despite a brief mention in the second paragraph. To better connect laboratory findings with clinical relevance, a more comprehensive discussion of animal model studies is warranted.

5) The manuscript currently lacks figures and tables, which reduces its accessibility and reader engagement. Consider adding: A central figure illustrating the proposed relationship between iron and VC. Tables summarizing: (1) in vitro evidence of iron's pro- and anti-calcific effects (including potential mechanisms), (2) in vivo evidence with corresponding mechanisms, and (3) clinical studies demonstrating the relationship between iron and VC. These additions would help readers gain a clearer understanding of the current knowledge and its limitations.

6) Chapter 3, “Treatment of Vascular Calcification Based on Iron Metabolic Pathways”, lacks sufficient clinical evidence specifically addressing VC. Much of the content focuses on the effects of iron supplementation on anemia or general clinical conditions, rather than its direct role in VC. This chapter would benefit from a major revision and reorganization. A suggested structure is 1) observational clinical evidence linking iron deficiency or overload to VC, and 2) the therapeutic potential of iron modulation in VC as demonstrated in clinical studies.

7) Numerous typos and terminology errors were found. Examples include Page 3, line 90, metabolic should be metabolism., Page 4, line 130, calpain and smoothin should be calponin and smoothelin., Page 5, line 170, the phrase “membrane-mediated cells” is unclear. A thorough proofreading is recommended to correct these and other errors throughout the manuscript.

8) The claim that malignant arrhythmia is directly associated with VC is questionable. Please provide specific evidence or references to support this statement, or reconsider its inclusion in the clinical context.

9) The citation style throughout the manuscript is inconsistent, and in some cases incorrect. For example, Page 5, line 165 (C. et al. 2015) should be (Zoccali C, et al. 2015). Page 6, line 199 (Aya et al. 2024) should be (Komuro A, et al. 2024). Please review all references and ensure they follow the appropriate format.

Experimental design

-

Validity of the findings

-

Reviewer 2 ·

Basic reporting

1. It would be beneficial to reorganize the findings on iron metabolism and VC into two separate tables: one summarizing key evidence from basic science research (e.g., cellular and animal models) and another dedicated to clinical and epidemiological studies. This would provide readers with a more structured, concise, and intuitive overview of the current landscape in this field.

Experimental design

1. The description of VC in the first section is not sufficiently aligned with the theme of iron metabolism. It is recommended to condense this part and focus more on the aspects related to iron metabolism.

2. The discussion regarding the dual role of iron metabolism in VC should be expanded. It is recommended to add subsections to improve the structural clarity, particularly concerning the impact of iron-dependent programmed cell death, ferroptosis, on VC.

Validity of the findings

1. The current discussion in the second section on the relationship between iron metabolism and VC reads somewhat like a list of individual study results, lacking a thorough synthesis of the evidence. It is recommended that the authors provide a more integrated analysis to highlight the connections and contradictions between different studies. Furthermore, the addition of a comprehensive mechanistic pathway figure would greatly help to visually summarize the proposed mechanisms and enhance the overall clarity for readers.

---

## Round 0.2 · accepted · Accept

· Academic Editor

Accept

All issues pointed by the reviewers were adequately addressed and the manuscript was substantially revised.